# Prediction Tool to Estimate Potassium Diet in Chronic Kidney Disease Patients Developed Using a Machine Learning Tool: The UniverSel Study

**DOI:** 10.3390/nu14122419

**Published:** 2022-06-10

**Authors:** Maelys Granal, Lydia Slimani, Nans Florens, Florence Sens, Caroline Pelletier, Romain Pszczolinski, Catherine Casiez, Emilie Kalbacher, Anne Jolivot, Laurence Dubourg, Sandrine Lemoine, Celine Pasian, Michel Ducher, Jean Pierre Fauvel

**Affiliations:** 1Hospices Civils de Lyon, Service de Néphrologie, Hôpital Edouard Herriot, Université Claude Bernard Lyon 1, CEDEX, F-69437 Lyon, France; maelys.granal@chu-lyon.fr (M.G.); lydia.slimani@chu-lyon.fr (L.S.); nans.florens@gmail.com (N.F.); florence.sens@chu-lyon.fr (F.S.); caroline.pelletier02@chu-lyon.fr (C.P.); romain.pszczolinski@chru-strasbourg.fr (R.P.); casiez_catherine@live.fr (C.C.); emilie.kalbacher@chu-lyon.fr (E.K.); anne.jolivot@chu-lyon.fr (A.J.); laurence.dubourg@chu-lyon.fr (L.D.); sandrine.lemoine01@chu-lyon.fr (S.L.); celine.pasian@chu-lyon.fr (C.P.); 2Pharmacie, Hospices Civils de Lyon, EMR3738 Ciblage Thérapeutique en Oncologie, Université Claude Bernard Lyon 1, CEDEX, F-69437 Lyon, France; michel.ducher@chu-lyon.fr

**Keywords:** potassium diet, prediction tool, Bayesian network, chronic kidney disease, Epidemiology

## Abstract

There is a need for a reliable and validated method to estimate dietary potassium intake in chronic kidney disease (CKD) patients to improve prevention of cardiovascular complications. This study aimed to develop a clinical tool to estimate potassium intake using 24-h urinary potassium excretion as a surrogate of dietary potassium intake in this high-risk population. Data of 375 adult CKD-patients routinely collecting their 24-h urine were included to develop a prediction tool to estimate potassium diet. The prediction tool was built from a random sample of 80% of patients and validated on the remaining 20%. The accuracy of the prediction tool to classify potassium diet in the three classes of potassium excretion was 74%. Surprisingly, the variables related to potassium consumption were more related to clinical characteristics and renal pathology than to the potassium content of the ingested food. Artificial intelligence allowed to develop an easy-to-use tool for estimating patients’ diets in clinical practice. After external validation, this tool could be extended to all CKD-patients for a better clinical and therapeutic management for the prevention of cardiovascular complications.

## 1. Introduction

Some modifiable lifestyle factors, such as diet, particularly potassium intake, may influence the risk of life-threatening cardiovascular events. Many studies have reported a relationship between increased dietary potassium intake and lower blood pressure, resulting in a decreased risk of cardiovascular (CV) events [1,2,3,4,5,6,7,8,9] but the relationship is likely to be a U-shaped curve. Indeed, if insufficient potassium intake is linked with an increased CV risk, an excessive intake of potassium could also be responsible for an increased risk of hypertension, stroke, and CV mortality [10,11]. The relationship between kalemia and CV risk has also been demonstrated with a higher risk in either hypokalemia or hyperkalemia [12,13,14,15]. However, a direct relationship between kalemia and potassium intake has not been demonstrated so far, even in renal failure. In order to prevent the onset of cardiovascular disease, the WHO in 2013 recommended, in the general population, a daily dietary intake of potassium higher than 90 mmol/day. In Europe, the average consumption of fruits and vegetables is too low to reach this recommendation [16].

Even if the trend is toward an increase of the dietary intake of potassium in the general population, excess can be dangerous for patients with CKD [10,11] whose potassium is no longer eliminated properly by the kidneys. However, the latest KDIGO guidelines [17] for the clinical management of CKD do not provide specific guidance on potassium intake but recommend that “people with CKD receive expert dietary advice and information as part of an education program tailored to the severity of CKD and the need for salt, phosphate and potassium intervention”. More recently, the KDOQI guidelines [18] in 2020, recommended the assessment of dietary potassium intake in stage 3 to 5 CKD patients, especially those with hypo- or hyperkalemia [19]. The KDOQI guidelines also recommend dietary potassium intake to be assessed using several complementary methods, such as dietary questionnaires and 24-h urine collection to measure the accuracy of dietary potassium intake estimates. In patients with CKD, 24-h urine collection for potassium measurement is a reliable method for measuring dietary potassium intake, but it is cumbersome and, therefore, not widely used in clinical practice. To overcome the 24-h urine collection, some trials tried to estimate the potassium-diet intake with spot urine collection and dietary questionnaires. Unfortunately, this method (estimation of 24-h urinary potassium excretion from a urine spot) failed to be reliable for individual estimates [20,21,22]. The KDOQI guidelines suggest that methods for assessing dietary intakes should be simplified to obtain reliable data on dietary intakes and ensure that they are culturally appropriate.

Most popular dietary surveys use 24-h dietary recalls, dietary records based on dietary databases and the food frequency questionnaires (FFQ) [23,24]. However, these survey-based methods suffer from lack of reliability [23]. These methods usually underestimate the actual dietary intake of potassium [25,26,27]. In CKD patients, this poor estimation may be related to parameters related to patient characteristics, pathology or treatment that is not usually captured by dietary questionnaires. Taken together, it seems essential for clinicians to have an easy-to-use and reliable tool for estimating potassium intake in CKD patients. In addition, because the impact of potassium intake is controversial in CKD patients, the development of an easy-to-use and reliable tool is of great interest in medical research to support future recommendations.

Recently, models involving artificial intelligence have proven their abilities to solve real problems, particularly in the health field, based on self-learning of databases [14,28,29,30,31]. These new approaches allow the creation of diagnostic or prognostic tools, leaning against existing rules and gold standards. This new approach could help the practitioner in his decisions. Thus, this study aimed to develop a clinical tool to estimate potassium intake using 24-h urinary potassium excretion as a surrogate in CKD patients using Bayesian networks derived from artificial intelligence. In addition, the developed tool evaluated the relative importance of demographic, therapeutic, nephrological and dietary variables in estimating potassium diet.

## 2. Materials and Methods

### 2.1. Patients and Data Pre-Processing

The study was offered to 540 adult patients attending the nephrological consultation of the Herriot Hospital (Lyon, France) between October 2019 and July 2021 and who routinely collected their 24-h urines for sodium and/or potassium 24-h urinary excretion determination. Exclusion criteria for the “potassium part” of the study were: (i) no urinary potassium excretion determination, (ii) medication that could influence 24-h kaliuresis (sodium polystyrene sulfonate, potassium supplementation), (iii) potassium loss through diarrhea, vomiting or heavy sweating (sports), (iv) non-reliable 24-h urinary collection, and (v) cardio-renal syndrome.

Urinary potassium excretion collected over a 24-h period was a surrogate for estimating dietary potassium intake. Urine creatinine measurement and patient interviews were used to assess the completeness of 24-h urine collection. The study was approved on the 26 September 2019 by the independent “Comité de protection des personnes OUEST II”.

### 2.2. “UniverSel” Self-Questionnaire

The “UniverSel” self-questionnaire was created with the help of a dietician to estimate the sodium/potassium consumption of patients followed in nephrology units. This self-administered questionnaire based on the self-administered questionnaire created by Dr. Robard Martin [32,33] collected: (i) the frequency of consumption of servings of vegetables, mushrooms, pulses and/or potatoes, fruits and/or fruit juices, bananas, dried fruits, chocolate, (ii) the size of meal servings determined by the patient himself/herself in relation to that of his/her relatives or friends, (iii) the patient’s characteristics: age, weight, height and gender, (iiii) the day and month of the urine collection

Patients were also questioned about the occurrence of the following events in the week prior to urine collection: diarrhea, vomiting or heavy sweating. In accordance with French regulatory authorities dedicated to non-interventional studies involving the human person, patients were asked if they did not object to the collection of their data in an anonymous and confidential manner for statistical analysis.

This self-questionnaire is available online [34].

### 2.3. Baseline Variables

Twenty-five variables concerning the patients’ characteristics (weight (Kg), height (m), age (year), gender (F/M), Systolic Blood Pressure (SBP) (mmHg), Diastolic Blood Pressure (DBP) (mmHg)), their comorbidities (original nephropathy (hypertension, diabetes, tubule interstitial, glomerular, autosomal dominant polycystic), oedemas (yes/no), heart failure (yes/no), nephrotic syndrome (yes/no)), their biological work-up (creatininemia (µmol/L), estimated glomerular filtration rate using the Chronic Kidney Disease EPIdemiology formula [35,36] (eGFR) (mL/min/1.73 m²), kalemia, plasma bicarbonate concentration, 24-h urine potassium excretion (24-h kaliuresis) (mmol/L)), their treatments (anti-hypertensive (0, 1, 2, 3 or more)) and/or diuretics (ARB or ACEI, Furosemide, Thiazides, spironolactone) (yes/no) and their answers to the UniverSel self-questionnaire (vegetables, fruits, bananas, mushrooms, chocolate, dried vegetables and/or potatoes, dried fruits, proportion of meals) were collected in order to estimate the 24-h kaliuresis taken as a reference of daily dietary potassium consumption. Quantitative variables (age, weight, height, eGFR, SBP, DBP) were discretized in five classes of equal frequencies. A database was created containing the data of the 25 variables. Ethnicity was also recorded to describe the included population. Diuresis and creatininuria were also recorded to ensure proper 24-h urine collection. The database was cleaned, missing data were handled by Bayesian imputation, and the data were formatted before the prediction tool was developed. The 24-h kaliuresis, was arbitrarily autodiscretized into three clinically relevant classes of identical frequency: less than 50 mmol/day, 50 to 69.9 mmol/day, and greater than 70 mmol/day.

### 2.4. Development and Optimization of the Prediction Tool

The prediction tool was developed using a Bayesian network issued from artificial intelligence. A Bayesian network is a directed acyclic graphical model composed of nodes and directed arrows. The nodes represent the variables and the directed arrows represent the probabilistic relationships between the variables. The basis of the Bayesian network is the conditional probability rule of Bayes’ theorem that defines each directed arrow. Bayesian networks take into account both a priori expert knowledge and the experience contained in the data. Thanks to artificial intelligence, they model the knowledge on the subject and try to reproduce the acquired reasoning on new queries.

A Bayesian Tree augmented Naive (TAN) network was used to connect clinical, therapeutic, biological and dietary parameters to estimate patients’ 24-h kaliuresis. The TAN algorithm applies three rules: (i) each node is independently linked to the target node (i.e., all-cause mortality); (ii) each node is also linked to a single parent node; (iii) among all possible structures, the one that maximizes the overall mutual information (mutual information measures the strength of the relationship between each variable and the target) between the nodes is chosen. The strengths of the relationship between the variables and the target (24-h kaliuresis) were evaluated by the percent variance of beliefs using Netica^®^ 5.19 software. The main variables with the highest percentage of variance were selected. Thus, to improve the ergonomics of the model and to make it usable in clinical practice, the number of variables was reduced to 14, allowing to preserve the performance of the prediction network. Patients were then randomly assigned to a training set (80%) and a validation set (20%). The training set was used to construct the prediction tool. Internal validation of the prediction tool verified the good fit (in terms of accuracy/error rate) between the predicted and observed 24-h kaliuresis class of the validation set.

### 2.5. Statistical Analysis 

Means ± standard deviation (SD) were calculated for each quantitative variable if it followed a normal distribution. Otherwise, values were expressed as median [Interquartile Range]. Discretized quantitative variables and qualitative variables were described in terms of percentages. The statistical software used for data analysis was Excel version 2016 using pre-existing formulas and BiostaTGV [37] website for online statistical testing.

## 3. Results

### 3.1. Study Population

The study was proposed to 540 adult patients who routinely collected their 24-h urines and who were followed in the nephrology department at the Edouard Herriot Hospital (Lyon, France) between October 2019 and July 2021. Among them, 483 patients had an exploitable 24-h urine collection (exclusion for non-exploitable 24-h urine collection suspected on 24-h creatininuria and confirmed by interview N = 43; treatment interfering with 24-h urine collection (tolvaptan) N = 3; food supplements N = 2; diarrhea N = 1; sports N = 2; No UniverSel questionnaire N = 6). Among them were 70 patients whose 24-h kaliuresis was not determined, 35 patients taking medication that interfered with K urinary excretion (sodium polystyrene sulfonate, potassium supplementation), 3 patients with a cardio-renal syndrome) were excluded. None of the patients refused to participate (Figure 1).

Finally, 375 patients were included (33.1% women, mean age ± SD was 64 ± 15 years; mean height ± SD was 1.68 ± 0.09 m and mean weight ± SD was 78.8 ± 15.9 kg). Most patients were Caucasian (87.7%). The mean ± SD eGFR of the subjects was 52.4 ± 23.9 mL/min/1.73 m². Healthy volunteers or stage I CKD patients represented 8.8% of included patients, 26.1% for stage II, 22.9% for stage IIIa, 24.5% for stage IIIb, 15.2% for stage IV and 2.4% for stage V.

Mean ± SD 24-h creatininuria was 12.0 ± 4.3 mmol/24h. The 24-h kaliuresis, used as a reference for potassium intake, was discretized into three classes of equivalent frequency. It was estimated that 34.9% of patients consumed less than 50 mmol per day, 32.5% of patients consumed between 50 and 69.9 mmol per day, and 32.5% consumed more than 70 mmol per day. Hypertensive nephropathy was the most frequent type of nephropathy (30.1%) in the study population, followed by diabetes (18.7%) and glomerular nephropathy (12.5%). The population main characteristics are listed in Table 1.

### 3.2. Variables Selected for the Development of the Clinical Prediction Tool and Internal Validation

Using a Bayesian network, the variables were first sorted according to the percent variance of beliefs (Table 2) to define the most informative variables. The 14 variables most correlated with the target variable (i.e., 24-h kaliuresis, stratified into three classes), were selected for the creation of the tool for estimating daily potassium consumption based on 24-h urinary potassium excretion of patients followed in nephrology. The Bayesian network used to create the tool for predicting potassium consumption based on 24-h urinary potassium excretion is illustrated in Figure 2.

In our cohort, the five most informative variables for estimating 24-h potassium excretion were, in descending order, weight, height, age, meal portion (subjectively determined) and eGFR. The 25 analyzed variables are sorted in descending order according to their percent of variance beliefs in Table 2.

The results of the internal validation to test the fit between the predicted and the observed 24-h kaliuresis class, to estimate dietary potassium intake, in the validation data set are given in Table 3. The accuracy of the prediction was 74%.

## 4. Discussion

Controlling potassium intake is critical in the management of CV disease among CKD patients. The potassium diet estimation tool proposed herein was developed in comparison to 24-h kaliuresis using a Bayesian network (machine learning model capable of self-learning from a training database); this prediction tool was accurate enough to be proposed in clinical practice.

Urinary potassium excretion may not necessarily represent dietary potassium intake in CKD patients since mechanisms involved in potassium homeostasis and excretion are impaired in patients with CKD. Nevertheless, in a stable chronic situation, kaliuresis remains the best indicator of potassium intake. Several studies used urinary potassium excretion as surrogate for dietary intake and a direct relationship between potassium excretion and CV or renal risk has not been demonstrated so far [18].

Among the 14 parameters used for its assessment, surprisingly, the three most correlated parameters with 24-h potassium excretion were weight, height and age (in descending order), which are physical characteristics of the patients. The “meal portion” item proposed by Robard-Martin [33] reflecting patients’ thoughts of eating “more”, “less” or “like” their relatives, was among the items retained in the final proposed tool. In our cohort, the frequency of intake of potassium-rich foods (fruits, vegetables, dried fruits, bananas, mushrooms, chocolate and dried vegetables) poorly correlated with the real amount of potassium in the diet. These results may explain the poor adequacy usually reported between dietary surveys and 24-h kaliuresis, which only consider the potassium-rich foods consumed by the patient. Our tool, which is reliable and easy to use, seems to be more suitable for estimating potassium consumption than a simple questionnaire collecting only patients’ dietary habits.

In CKD patients included in the UniverSel study, parameters of nephrological interest (blood pressure, glomerular filtration rate, type of kidney disease, plasma bicarbonate concentration) were found to be the most influent on subjects′ kaliuresis (Table 1). Lower eGFR, plasma acidosis, elevated blood pressure and hypertensive nephropathy were most related to a lower 24-h urinary potassium excretion. Other diuretics (furosemide, thiazides) chronically ingested as a therapeutic treatment did not influence kaliuresis and were not retained in the finally proposed prediction tool. Based on these parameters, the nephrologists and dieticians are likely to modulate their advice regarding potassium intake and adjust their treatment, taking into account the severity of the chronic renal failure. The proposed tool for assessing dietary potassium intake using 24h urinary potassium excretion has a satisfactory internal performance. Indeed, the accuracy of the model is 74% which means that the percentage of predicted and observed subjects in the same consumption potassium class is 74%. The prediction tool developed with the help of a Bayesian network derived from artificial intelligence is therefore suitable for use in clinical practice.

This study brings a real originality since, to our knowledge, no Bayesian model has been used to develop a tool to estimate food diet. The most used prediction tools are usually built from linear regression, which does not consider non-linear factors, which is often the case in health science. Moreover, Bayesian networks are well adapted to the analysis of categorical data recorded in diet questionnaires. This statistical method is also well adapted for managing data out of filled questionnaires. Indeed, Bayesian imputation could easily manage missing data by attributing the most highly probable value based on the main criterion of judgment and other patient’s characteristics.

However, this study has some limitations. Potassium consumption was estimated by a single 24-h kaliuresis, taken as a reference, and a 3-day kaliuresis would have been more accurate. Nevertheless, most of the included CKD patients were accustomed to collecting their 24-h urine annually (91.5%), mitigating the risk of having an abnormal collection in our dataset. Moreover, the clinician systematically questioned the patient about the conditions of collection (non-habitual meals, missing urine, etc.). Thus, forty-three 24-h urine collections were discarded based on clinical interview. Exclusion of these abnormal conditions of collection allowed to have a reliable and robust tool. On average, the mean 24-h creatinuria reported in this study (12.0 ± 4.3 mmol/24 h) reflects the completeness of the 24-h urine collections. The monocentric design of our study limits the direct extrapolation of the proposed tool to a wider use. However, our study had the same frequencies of pathologies than reported by the national REIN registry (hypertensive nephropathy: 25%, diabetic nephropathy: 21%, glomerular nephropathy: 10%, polycystic fibrosis: 5.5%) [38]. This argues for a good sampling of our population. This could be explained by the large number of patients included who were followed chronically by six different physicians and by the great diversity of recruitment in our university hospital. Nevertheless, lack of external validation represents the main limitation of our study. To ensure its integration in clinical practice, the proposed tool is now tested in two European centers that prospectively record the variables used in our tool in comparison with 24-h kaliuresis. In the medium term, if the validation obtains satisfactory results, the data will be integrated into our learning database to increase sample size in accordance with the artificial intelligence used to build our tool.

After validation, this estimation tool for potassium consumption will be proposed as an ergonomic and user-friendly application to allow users (CKD patients) to evaluate their potassium consumption and increase their compliance to the dietary recommendations proposed by health professionals. The tool can also be integrated into an existing medical application dedicated to CKD patients’ health (therapeutic compliance, blood pressure and remote monitoring). This tool could also be integrated into the clinical routine of various healthcare professionals involved in nephrology follow up (nephrologist, general practitioner, nurse practitioner and dietician). By allowing an easy control of potassium consumption, this tool may help in medical and dietary decision-making and help for adapting the patient’s therapeutic management. Finally, this tool could be included in global strategies to reduce CV morbidity and mortality, which remains the main burden in CKD patients.

## 5. Conclusions

This study proposes a tool for estimating 24-h urinary potassium excretion in CKD patients who are at high risk of CV mortality. The internal validation of the tool resulted in a very satisfactory performance with an accuracy of 74%. In an original way, the variables related to potassium consumption were more related to clinical characteristics and renal pathology than to the potassium content of the ingested food. The originality of the study also lies in the use of a method derived from artificial intelligence techniques to develop a tool for estimating patients’ diets. External validation is required for this proposed tool before using it in clinical practice.

## Figures and Tables

**Figure 1 nutrients-14-02419-f001:**
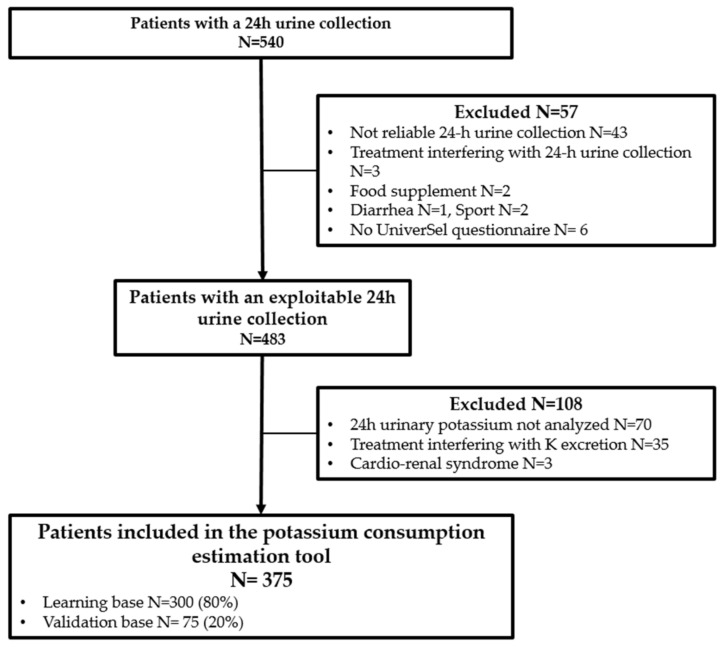
Flow chart of the UniverSel study population.

**Figure 2 nutrients-14-02419-f002:**
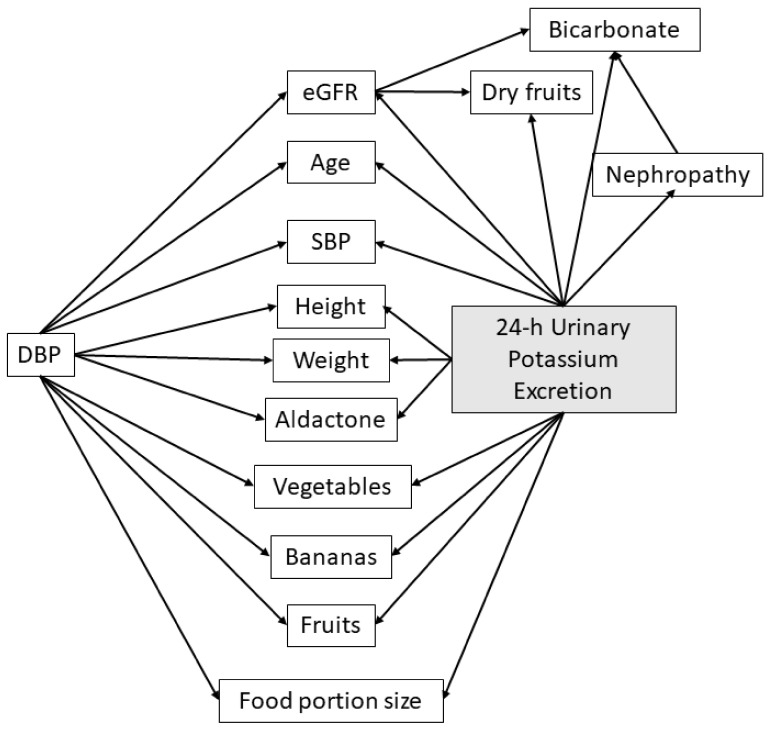
Optimized Bayesian network structure of the tool for estimating 24-h urinary potassium excretion. Abbreviations: eGFR—estimated glomerular filtration rate; DBP—Diastolic Blood Pressure; Nephropathy—nature of the nephropathy; SBP—Systolic Blood Pressure.

**Table 1 nutrients-14-02419-t001:** Characteristics of CKD patients included in the UniverSel study.

**24-h Potassium Urinary Excretion**	**Mean**	**SD**	**Distribution (%)**
Less than 50 mmol/day	38.6	8.8	34.93
50 to 69.9 mmol/day	58.8	5.6	32.53
More than 70 mmol/day	89.5	18.5	32.53
**Patient characteristics**	**Mean**	**SD**	**Distribution (%)**
Gender			
M			66.9
F			33.1
Age (years)	64	15	
Weight (kg)	78.8	15.9	
Height (m)	1.68	0.09	
Nephropathy			
Hypertension			32.0
Diabetes			18.7
Tubulo interstitial			16.8
Glomerular			12.5
Autosomal Dominant Polycystic			5.3
Other			14.7
CKD stage			
I (≥90 mL/min/1.73 m²)			8.8
II (60–89 mL/min/1.73 m²)			26.1
IIIa (45–59 mL/min/1.73 m²)			22.9
IIIb (30–44 mL/min/1.73 m²)			24.5
IV (15–29 mL/min/1.73 m²)			15.2
V (<15 mL/min/1.73 m²)			2.4
SBP (mmHg)	133.5	16.4	
DBP (mmHg)	75.1	11.7	
Number of antihypertensive drugs			
0			16
1			22.1
2			26.1
3 or more			35.7
Diuretics (Yes)			37.3
Oedema (Yes)			8.7
Diabetes (Yes)			26.7
Heart failure			9.1
Ethnic origin			
African			10.4
Caucasian			87.7
Asian			1.9
Month of inclusion			
January			8
February			7.5
March			12.3
April			3.5
May			10.9
June			18.4
July			9.3
August			3.7
September			7.7
October			6.4
November			8
December			4.3
**Biology**	**Mean**	**SD**	
eGFR (ml/min/1.73 m²)	52.4	23.9	
Kalemia (mmol/L)	4.4	0.5	
Bicarbonates (mmol/L)	25.5	2.9	
Creatinemia (µmol/L)	140.5	69.9	
24-h diuresis (L/day)	1.9	0.6	
24-h kaliuresis (mmol/day)	61.7	24.3	
24-h creatinuria (mmol/day)	12.0	4.3	

Abbreviation: DBP—Diastolic Blood Pressure; eGFR—estimated glomerular filtration rate; F—Female; M—Male; Nephropathy—nature of the nephropathy; SBP—Systolic Blood Pressure.

**Table 2 nutrients-14-02419-t002:** Percentage variance between explanatory variable and potassium consumption for all 25 baseline variables.

Variables		Percentage Variance of Beliefs
Variables included in the optimized Bayesian network		
1	Weight	4.91
2	Height	4.66
3	Age	4.02
4	Food portion size	3.18
5	eGFR	2.8
6	Nephropathy	2.39
7	Fruits	1.9
8	Spironolactone	1.37
9	Diastolic blood pressure	1.37
10	Vegetables	0.94
11	Bicarbonate	0.74
12	Systolic blood pressure	0.74
13	Dry Fruits	0.74
14	Bananas	0.64
Variables not included in the optimized Bayesian network		
15	Gender	0.49
16	Oedema	0.49
17	Mushrooms	0.43
18	Kalemia	0.32
19	Heart Failure	0.31
20	Nephrotic syndrome	0.27
21	Chocolate	0.25
22	Thiazides	0.23
23	Furosemide	0.22
24	Renin angiotensine sytem blockers	0.15
25	Dry vegetables	0.07

Abbreviation: eGFR—estimated glomerular filtration rate.

**Table 3 nutrients-14-02419-t003:** Table of agreement between the predicted and observed 24-h urinary potassium excretion used to estimate dietary potassium intake of the 375 patients analyzed in the UniverSel study.

		Estimated 24-h Kaliuresis		
		Less Than 50 mmol/day	From 50 to 69.9 mmol/day	More Than 70 mmol/day
Observed 24-h Kaliuresis	Less than 50 mmol/day	85 (70%)	20	17
	From 50 to 69.9 mmol/day	17	96 (73%)	18
	More than 70 mmol/day	16	10	96 (79%)

The number of patients (also expressed as a percentage) whose 24-h urinary potassium excretion was correctly predicted for each category of observed 24-h urinary potassium excretion is shown in the dark square in the diagonal of the table.

## Data Availability

All databases are protected in a password-protected Excel file and stored on password-protected computers. The passwords are changed every 3 months. The databases will be destroyed in 20 years.

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
