# Peer review of "Prediction Tool to Estimate Potassium Diet in Chronic Kidney Disease Patients Developed Using a Machine Learning Tool: The UniverSel Study"

_nutrients, 2022, doi:10.3390/nu14122419_

Round 1
Reviewer 1 Report
General comments:
This study sought to develop a new tool for estimating potassium intake in patients with CKD using artificial intelligence. Given that all current methods of assessing potassium intake have notable limitations, this topic is timely and relevant, and the use of artificial intelligence is novel. However, the paper would benefit from some revisions and additional considerations, as described below.
Specific Comments:
Lines 35-37: This point is not adequately addressed or explained, and the only reference provided here that actually supports that point is reference 3.
Lines 41-42: High BP is also one of the primary causes of CKD.
Lines 46-47: You suggest here that CKD patients should be limiting their potassium intake, but this is on the higher end of the WHO recommendations cited earlier. In addition, the KDIGO guideline cited is outdated; guidelines for clinical management of CKD don’t provide specific potassium intake guidelines: https://kdigo.org/wp-content/uploads/2017/02/KDIGO_2012_CKD_GL.pdf. They only provide specific recommendations for sodium and protein intake. In addition, KDIGO guidelines for BP management in CKD only recommend avoiding potassium-containing salt substitutes and high potassium diets in advanced CKD, without providing a specific potassium recommendation: https://kdigo.org/wp-content/uploads/2016/10/KDIGO-2021-BP-GL.pdf. Overall the evidence suggesting potassium restriction in CKD is controversial, at best, especially when examining food versus supplements as well as early stages versus advanced CKD.
The intro, and perhaps the discussion, should incorporate the nuances in the literature regarding the relationship between potassium and CKD progression/complications. I would argue that the best justification for needing this new method of estimating potassium intake is not for clinical monitoring of potassium intake in CKD because we “know” it is bad, but rather to advance the research in this field to continue studying the relationship between potassium intake and CKD outcomes, potassium and BP in CKD, etc. so more accurate and appropriate guidelines can be developed.
Lines 51-52: While this is true for healthy adults, evidence suggests that urinary potassium excretion decreases as CKD progresses (https://pubmed.ncbi.nlm.nih.gov/27366661/#:~:text=Results%3A%20Urinary%20potassium%20excretion%20gradually,%25%2C%20P%20%3C%200.001), which makes sense given the decline in kidney function and other metabolic and hormonal changes that occur.
Lines 88-95: How can you justify using this questionnaire when you mention in the intro how inaccurate self-report is?
Table 1: It would also be helpful to include the range of eGFR, could also be in text. That would help readers understand the range of CKD stages included in the sample. Maybe also include serum BUN and creatinine levels.
Lines 177 and 178: It is important to be clear throughout that you are predicting 24-hour urinary potassium excretion, not actual intake. Though you are using 24-hour urinary potassium as your indicator of intake, it is not potassium intake. This verbiage should be clear and specific throughout.
Line 179: Again, why include a subjective variable at all if it is not reliable?
Line 197: Change title to remove intake and say 24-hour urinary potassium excretion. Also, what exactly are these agreement values in the table telling us? More details in the text or as a table footnote should be provided, especially for readers who may not be as familiar with this method.
Either in the text or in a table, is it also possible to provide the actual, average predicted 24-hour potassium excretion values along with the measured 24-hour potassium excretion values, and the mean difference? More details about these results are needed.
Lines 223-225: But your tool still includes these inaccurate measurements…
Lines 228-230: Perhaps they were associated with lower potassium excretion due to disease progression, but not necessarily a lower potassium diet. See previous comment regarding the accuracy of 24-hour urinary potassium excretion in CKD.
Line 262: missing reference
Lines 265-282: What if the external validation results are not satisfactory, and thus the tool is not validated? This section of the discussion seems to imply that validation is assumed.
Reviewer 2 Report
Dear Author/s,
Here are my some comments/feedback for your manuscript to increase the quality of your work.
Introduction:
Line 65: Include reference.
Material and Methods:
Line 74: What do you mean by adult patients? Are they suffering from CKD? If so, which stages of CKD and what was the proportion of different stages of CKD recruited patients? If you included CKD patients, how did you define CKD and how did you do staging of CKD?
Line 76-77: This part needs to be rephrased. What do you mean by inclusion criteria based on the determination of 24 hours potassium? Is it about the cutoff point of potassium? If so, please mention how did you include in the study?
Line 80: How did you define reliable vs. non-reliable 24 hours criteria? Please mention in detail. This is important for readers.
Line 81: You don't need to include this point. Remove this point.
Line 86: This is not clear. Rephrase them. Is it project number quote or typing errors?
Line 92: Include reference.
Line 98: Define clearly who is regulatory authorities.
Line 103 and 104: Mention their unit in parenthesis after each parameter.
Line 106 and 107: You need to include reference for this calculation.
Line 119 and 120: What is the reference of these three classes? If it is your own classification, you need to state the rational for taking this cut-off point for the classification.
Line 128 and 129: This sentence looks fancy in this paragraph. You may switch to acknowledgment section this sentence.
Line 145-149: This statistical analysis section is not satisfactory and not sufficient. Which statistical software used for the analysis of the data to show mean+/-SD? Mention detial about the version and product of the that software you used for that purpose.
Why did you show the data as Mean+/-SD? why not Mean+/-SEM?
Results:
Line 152-154: Mention what stage of CKD or normal person for just check up?
Line 163-164: Check the structure of this sentence.
Line 189 or table 1: What is plasma potassium concentration?
What is the relationship between the plasma potassium concentration vs. urinary excretion? is it inverse?
does the excretion of potassium have the impact on plasma sodium concentration and urinary sodium excretion?
What is the excretion of sodium with these grades of potassium excretion?
Does the potassium excretion and sodium excretion correlate to each other?
Plasma level of sodium and urinary excretion of sodium cannot be excluded since they are also involved in secondary hypertension and cardiac outcomes.
What is the anionic gap with this potassium urinary excretion?
Do the potassium excretion and anionic gap correlate?
Table 2: Check spelling: Percentage, Syndrome, etc.
Line 197-198 or Table 3: Only this table is not sufficient. Please respond the following questions.
- what is the approximate 24-hour potassium intake, plasma level of potassium in these three groups of urinary potassium excretion?
- What is the odds ratio for the three different classes of urinary potassium excretion with the stages of CKD?
- Is there significant changes plasma concentration and urinary excretion with the intake of potassium?
- What is the impact of potassium intake, plasma potassium level and urinary excretion of potassium on plasma sodium level and urinary sodium excretion?
Line 201-203 or figure 2: This picture is not good enough to read. Please save this picture in at least of 300dpi and TIF file so that I can read and make a comment.
Increase the font size and use whole single page for figure only. Keep the figure legend in the next page. Otherwise, it is not readable.
For the picture; 1. use whole page, 2. increase font size inside the box, 3. save picture with 300dpi at least and TIF file.
Discussion:
Line 228-230: Is this statement from your study or from the literature? if it is from literature, cite it? if it is from your study, please show the data in the table.
Line 230-232: I could not see this correlation data in the table. please highlight this where you presented this data to claim this statement?
References or Line 312: References are not consistent and harmonious. Please be consistent and harmonious to a reference section. Please use the journal guidelines for the reference.
Best wishes,
BP

Round 2
Reviewer 1 Report
Comments have been sufficiently addressed.
Reviewer 2 Report
Dear Authors,
Thank you for including and addressing my comments. There are still few typho and grammatical errors, which are minor but you can correct.